# Ramie Yield Estimation Based on UAV RGB Images

**DOI:** 10.3390/s21020669

**Published:** 2021-01-19

**Authors:** Hongyu Fu, Chufeng Wang, Guoxian Cui, Wei She, Liang Zhao

**Affiliations:** 1Ramie Research Institute of Hunan Agricultural University, College of Agricultural, Hunan Agricultural University, Changsha 410128, China; hongyu_ffu@163.com (H.F.); clregina@163.com (W.S.); zhaoliang4203@163.com (L.Z.); 2Macro Agriculture Research Institute, College of Resource and Environment, Huazhong Agricultural University, 1 Shizishan Street, Wuhan 430000, China; cfwang@webmail.hzau.edu.cn

**Keywords:** ramie, yield estimation, RGB images, deep learning

## Abstract

Timely and accurate crop growth monitoring and yield estimation are important for field management. The traditional sampling method used for estimation of ramie yield is destructive. Thus, this study proposed a new method for estimating ramie yield based on field phenotypic data obtained from unmanned aerial vehicle (UAV) images. A UAV platform carrying RGB cameras was employed to collect ramie canopy images during the whole growth period. The vegetation indices (VIs), plant number, and plant height were extracted from UAV-based images, and then, these data were incorporated to establish yield estimation model. Among all of the UAV-based image data, we found that the structure features (plant number and plant height) could better reflect the ramie yield than the spectral features, and in structure features, the plant number was found to be the most useful index to monitor the yield, with a correlation coefficient of 0.6. By fusing multiple characteristic parameters, the yield estimation model based on the multiple linear regression was obviously more accurate than the stepwise linear regression model, with a determination coefficient of 0.66 and a relative root mean square error of 1.592 kg. Our study reveals that it is feasible to monitor crop growth based on UAV images and that the fusion of phenotypic data can improve the accuracy of yield estimations.

## 1. Introduction

Crop yield estimation is an important research content of precision agriculture, and it plays a crucial role in making decisions for agricultural production input and realizing accurate agricultural operation and management [1,2,3]. Crop yield estimation is also required in market risk management and policy adjustment [4]. The conventional sampling method for crop yield estimation is destructive and labor-consuming, with low accuracy; thus, it cannot meet the demands of modern agriculture. The establishment of a crop yield estimation model based on phenotypic characteristics to predict a crop yield has significantly improved the accuracy of crop yield estimations [5,6].

Remote sensing is now regarded as the best technology for monitoring and estimating phenotypic characteristics over large areas [7,8]. There are many remote-sensing platforms, such as satellites, aircrafts, and unmanned aerial vehicles (UAV). Low-altitude remote sensing using UAVs with multiple advantages, such as flexibility, nondestructive monitoring, low costs, and high throughput, has been increasingly used in the field of precision agriculture in recent years [9,10]. UAVs can carry different kinds of sensors, such as a high spectrometer, multi-spectrometer, and radar. However, the high price of these sensors prevents remote-sensing technology from being widely applied to agriculture. As a cheap sensing device, an RGB camera is becoming a promising tool for continuous observation [11,12].

Some advances have been made in crop phenotypic characteristic (nitrogen level [13], Leaf Area Index (LAI) [14,15], plant height [16,17], etc.) monitoring by using UAV-RGB remote sensing. Plant height information can be obtained from the crop surface model, which is created by using structure-from-motion techniques. Chang et al. (2017) reported that the relative root mean square error (RMSE) between the plant height and calculated value was 0.33 m [18]. Bendig et al. (2015) obtained the plant height information of summer wheat, and they found that the R^2^ of the plant height model was between 0.80 and 0.82 [19]. Some sample vegetation indices (VIs) such as Visible Atmospherically Resistant Index (VARI) [20], Normalized Difference Yellowness Index (NDYI) [21], Normalised Green-Red Difference Index (NGRDI) [22], and Green Leaf Algorithm (GLA) [23] were calculated based on images acquired from the visible wavelengths. These VIs can be used to estimate crop yields. The plant number is an important indicator for crop growth status and yield potential. At present, the method based on target detection is mainly used for crop plant count [24,25,26]. However, plant counting using target detection technology based on UAV images poses a great challenge due to limited spatial resolution, small object size, and complex background features [27]. Previous studies have confirmed that combining vegetation indices, plant height, canopy cover, plant density, and other multiple features can improve the yield estimation. A yield prediction model was established by integrating the common spectral vegetation indices and plant height, with a determination coefficient of 0.88 [28].

Recently, crop yield estimations based on phenotypic information obtained from images have greatly advanced. An artificial neural network (ANN) algorithm was used for the corn grain yield estimation based on the vegetation indices, canopy cover, and plant density [29]. Random forest and stepwise multiple linear regression were used to estimate the high-density biomass [30].

Ramie is one of the characteristic economic crops and fiber crops, and its industrial development is of great significance for optimizing the resource structure of the textile industry and promoting economic growth. At present, the ramie industry is facing great crises, and the absolute supply of ramie in the world is very scarce. The production of ramie cannot meet the market demand. The traditional ramie yield estimation relies on the experience of farmers, and the estimation accuracy is low; therefore, this traditional yield estimation is less practical and feasible. Since ramie is a crop with a relatively small planting area, there are few studies of the establishment of ramie yield estimation models. So far, no ramie yield estimation model based on phenotypic characteristics acquired from an UAV-RGB remote-sensing system is available.

In this study, the ramie yield under different doses of nitrogen fertilization was estimated based on RGB images obtained from sensors mounted on an UAV. The images were processed to extract information on the vegetation indices, plant heights, and plant numbers. The main objective of this study is to apply UAV-based spectral and structural information for the ramie yield estimation. This study will propose a new method for ramie plant counting so as to predict ramie yields by combining the plant number, plant height, and vegetation indices.

## 2. Materials and Methods

### 2.1. Study Area

The study area, located at the experimental base of Hunan Agricultural University, (28°11′01.981″ N, 113°04′10.159″ E) in Changsha, Hunan, approximately lies in the middle of the Yangtze River basin, and this area has a typical subtropical monsoon humid climate (Figure 1a). With abundant precipitation, sufficient light, and heat conditions, the study site is one of the main ramie production areas. Meanwhile, the flat terrain provides favorable conditions for UAV-based remote-sensing monitoring of ramie. Ramie is a perennial crop and can be harvested three times a year, so we did not need to replant it. In this experiment, the conventional ramie varieties Xiangzhu No. 3 and Xiangzhu No. 7 were used as materials. Each of two varieties was grown in 12 plots (a total of 24 plots), and each plot area was about 12 m^2^, with 4 rows × 8 columns. The planting density was 92,500 plants per hectare, and the row spacing was 0.2 m (Figure 1b). Four different nitrogen fertilizer treatments: 0 kg/ha (N0), 15 kg/ha (N15), 22.5 kg/ha (N22.5), and 45 kg/ha (N45) were randomly assigned with three replicates (Figure 1d). Nitrogen fertilizer were applied on 9 June 2019, when ramie was in the seedling stage. The N fertilizer application levels for each plot are shown in Figure 1c.

### 2.2. UAV Image Acquisition

In this study, we employed an Inspire 2 UAV (DJI, Shenzhen, China) with a maximum payload of 810 g and a flight duration of about 25 min in a windless environment. The platform carried a Zenmusex5S 35-mm HD digital camera (DJI, Shenzhen, China) with an effective pixel number of 20.8 million and a maximum resolution of 5280 × 3956. During the period of the second season of ramie in 2019 (Table 1), field images were acquired at 12:00–14:00 local time under clear and sunny weather conditions. Fifty-five images with pixels of 5280 × 3956 were obtained in each mission, and these obtained images were saved in 24-bit TIFF format.

The UAV platform was used to implement a predefined flight plan that was created in advance by a ground control station called DJI GS PRO. During the flight, the flight height was set as 18 m with a forward overlap of 80%, a side overlap of 70%, and a sensor angle of 90°; the exposure mode of the camera was automatically selected. The time of exposure was 1/200 s, ISO sensitivity was 100, aperture was 1.53, and flight speed was 1.1 m/s.

### 2.3. Image Processing

The key to yield prediction based on UAV images is to extract useful crop phenotypic information from the images. Therefore, prior to estimating the ramie growth and yield, image processing was implemented to obtain a set of image data, including VIs, plant height, and plant numbers.

#### 2.3.1. Image Mosaicking

Image mosaicking was accomplished by Pix4Dmapper (Lausanne, Switzerland) automatically. The basic operation process included: (1) checking POS data extracted from UAV images to ensure the anastomosis of longitude, latitude, altitude, heading angle, and swing angle; (2) screening all the images containing a certain point M, extracting the spatial position and attitude information of point M, restoring point M, and generating the point cloud map of the study area; and (3) adding the three-dimensional space information of the control points to the empty tri-ray editor to ensure the spliced images obtained at different time points were in the same coordinate positioning system. Finally, orthophoto images of ramie experimental area were generated based on the three-dimensional position of ground control points (GCPs).

#### 2.3.2. Vegetation Index Calculation

Crop canopy images obtained by UAV inevitably contain non-crop information such as shadow and ground. In order to avoid this interference information, the HSV (Hue, Saturation, and Value) channel was selected for soil segmentation. The averaged digital number (DN) values of the ramie canopy were obtained based on the segmentation images. In order to investigate the influence of the N levels and to select the most sensitive parameters, some VIs widely used in crop research were calculated. Table 2 presents the VIs selected in this study, their calculations, and the corresponding sources. The VIs included GLA, the excess red index (ExR), the excess green index (ExG), the excess green minus excess red index (ExGR), water index (WI), the normalised green-red difference index (NGRDI), the red green blue VI (RGBVI), and VARI.In addition, some digital variables were also used to reflect the growth trend of ramie, including g/r, g/b, and r/b.

#### 2.3.3. Plant Height Estimation

The calculation of the plant height was carried out in ArcGIS map software. Firstly, the digital surface model (DSM) and the digital terrain model (DTM) were constructed based on the acquired images at the seedling stage and maturing stage, respectively. DSM represented the absolute height of the crop canopy, while DTM represented the absolute height of the ground. Secondly, crop surface models (CSMs) were obtained by subtracting DTM from DSM with a raster math tool in ArcGIS. Subsequently, the average plant height of each plot was calculated. It was worth noting that 4 fixed ground control points were set in the experimental area, and the accurate coordinate information of these 4 control points was obtained in order to ensure the accuracy of the CSMs. Finally, the plant height calculated according to the UAV images was validated based on the field-measured plant heights.

#### 2.3.4. Plant Counting

The growth point of ramie was set as the target, and target detection algorithm based on the YOLOv4 model was used to count ramie in the canopy images. The YOLO target detection algorithm proposed by Redmon in 2016 was a continuation of GoogleNet, and it realized an end-to-end target detection algorithm. Subsequently, the YOLO algorithm continues to be improved, and YOLOv4 was created with increased precision, speed, and smaller target detection and with USES CSPDarknet53 as the Backbone, SPP as the additional Neck module, PANet as a characteristic fusion module of the Neck, and YOLOv3 as the Head. In this study, Yolov4.conv.137 was adopted as the pretraining model. During training, the momentum term was set as 0.949, the attenuation parameter was set as 0.0005, the initial learning rate was set as 0.001, and the batch size was set as 64.

The test platform was equipped with an Intel Core I9 9960X CPU, 4.0 GHZ master frequency, 16-core 32-thread processor, NVIDIA RTX 2080 GPU, 8 GB video memory, 32 GB DDR4 4000 MHz running memory, and Windows 10 Professional workstation edition and CUDA 10.2, CUDNN 10.2 software environment. Labeling was used to mark the growth point of ramie in the image. Sixteen images were randomly selected as the test set.

### 2.4. Field Date Collection

Measured field data at the maturity period, including plant height and yield. A scaled ruler was used to measure the shortest distance between the top of the ramie main stem and the ground. We took the average of 15 measured values as the field-measured plant height in the plot. An electronic scale was used to measure the actual ramie yield of each plot.

### 2.5. Model Construction and Statistical Analyses

In this study, the multiple linear regression model and the stepwise linear regression model were constructed, respectively. Microsoft Excel 2013 was used to construct the multiple linear regression model, and the stepwise linear regression model was constructed by IBM SPSS Statistics 22.0.0.0 (Armonk, New York, NY, USA). The statistical analyses, including the correlation analysis and regression analysis, were carried out using Microsoft Excel 2013 and IBM SPSS Statistics 22.0.0.0. The average plant height of each plot obtained from CSM was validated in reference to the ground-measured plant height, and the result was presented as a scatter plot. The correlation between the image feature data and yield was examined, and then, multiple linear regression and stepwise regression were, respectively, used to construct the yield estimation model of ramie, and the determination coefficients of the resultant models were evaluated. The obtained regression model was applied to the dataset validation. The robustness of the model was evaluated by analyzing the root mean square error (RMSE), relative error (RE), and standard error (SE).

## 3. Result

### 3.1. Effects of Different Nitrogen Levels

#### 3.1.1. Effect of Different N Levels on Plant Height

To explore the impacts of different N levels on the ramie plant height, an UAV monitoring (10 June) was carried out from the ramie second season seedling throughout the complete growth period of ramie. Figure 2 shows the effects of different N levels on plant height during the whole growth period. The growth cycle of ramie in the second season was about 46 days. During this period, the variation range of ramie plant height was within about 0–2.5 m, and the plant height in the first 20 days reached half of the whole plant height. This period (the first 20 days) was the key period to promote the growth of ramie plant height.

After the first 20 days, the difference in ramie plant height under different N levels began to increase. The plant heights of both ramie varieties at four different N levels ranked as follows: N15 > N45 > N22.5 > N0. The ramie plant height without any fertilizer application was the minimum. The ramie treated with N15 exhibited the maximum plant height. The results indicated that the ramie plant heights were related to the N levels, and they could be increased by applying an appropriate amount of nitrogen fertilizer; therefore, it is necessary to monitor ramie growth under different N treatments so as to improve the N fertilizer utilization efficiency in the field.

#### 3.1.2. Nitrogen Level Contributions to Yield

Figure 3 shows the effects of different nitrogen levels on the ramie yields. The differences in ramie yields of the same variety at the four N levels were not significant, but the differences in ramie yields of the two different varieties at different N levels were observed. For the variety Xiangzhu No. 3, the ramie yields at the four N levels ranked as follows: N45 > N15 > N22.5 > N0, while, for variety Xiangzhu No. 7, the ranking was N15 > N45 > N22.5 > N0. In general, the ramie yield in the plot without nitrogen fertilizer treatment was lower than that with N treatments. Therefore, it could be concluded that ramie yields could be increased by applying a certain amount of nitrogen fertilizer, which was consistent with a previous report that the yield of ramie was closely related to the N level. In this study, only one year of nitrogen treatment was observed, which might explain why the contribution of N application was not obvious for the same variety. It is necessary to monitor the growth of ramie under different N treatments for a longer term so as to improve the nitrogen efficiency and final yield.

### 3.2. Estimation of Ramie Yield Using UAV-Based Image Data

#### 3.2.1. Estimation of Plant Height and Plant Number Using UAV-Based Image Data

The plant height and plant number were calculated by using UAV-based image data, and the results with R^2^ and RMSE are shown in Figure 4. The determination coefficient of the plant height estimation model was 0.9095, and its RMSE was 0.052 m. The fitting degree of the plant number estimation model was also desirable (R^2^ = 0.9647, RMSE = 6.151). Our results indicated that the UAV-based image data could reflect well the actual ground measurement values.

#### 3.2.2. Relationship between Plant Height, Plant Number, and Ramie Yield

In order to explore the potential of reflecting the ramie yield using UAV-based structural features (plant height and plant number), a correlation analysis was performed. Table 3 shows the results of a correlation analysis between the UAV-based structural features and ramie yield. An extremely significant correlation between the UAV-based plant number and ramie yield was observed, with a correlation coefficient of 0.6. Since ramie is a crop whose yield is determined by its stalks, the plant heights were examined. The results indicated a significant correlation between the UAV-based plant heights and yields, with a correlation coefficient of 0.524. Taken together, the plant height and plant number information derived from the UAV-based images well-described the ramie yields; thus, these two indices could be used as input factors to establish a ramie yield estimation model.

#### 3.2.3. Relationship between VIs and Ramie Yield

Table 4 shows the results of the correlation analysis between the UAV-based spectral features and ramie yield. Our results indicated that the correlation relationship between the ramie yield and all of the visible spectrum indices calculated based on the UAV images was weak, of which WI, ExR, and VARI were positively correlated with the ramie yield, while NGRDI, NDYI, GLA, ExG, ExGR, G/R, G/B, R/B, and RGBVI were negatively correlated with the ramie yield. Among all the VIs, WI was the most significantly correlated with the ramie yield, and the correlation coefficient was 0.365. This result might be due to the fact that WI is a water-sensitive vegetation index. When the canopy leaves of ramie absorbed short-wave infrared radiation, they had a high moisture content. Therefore, WI could be an important indicator of crop health status and yield prediction. Our results also revealed a high correlation coefficient among the various VIs. The reason for this phenomenon might be that all VIs were obtained through mathematical transformation based on the digital number (DN) value extracted from the UAV-based images. Therefore, in order to avoid the saturation of spectral data and to increase the accuracy of the model, appropriate spectral parameters should be selected according to different crops, different varieties, and different growth periods during the construction of the crop yield estimate model.

#### 3.2.4. Estimation of Ramie Yield Using UAV-Based Image Data

The ramie yield estimation results based on the UAV images are shown in Figure 5a,b, represented by the stepwise linear regression results. In Figure 5a, the plant number was the input factor. In Figure 5b, the plant number and plant height were the input factors. Figure 5c shows the multiple linear regression results. As shown in Figure 5a, an R^2^ of 0.36 and RMES of 1.473 kg were obtained by fusing the plant numbers. As shown in Figure 5b, an R^2^ of 0.49 and RMES of 1.315 kg were obtained by fusing the plant heights and plant numbers. As shown in Figure 5c, the optimal yield estimation (R^2^ = 0.66, RMSE = 1.592 kg) was achieved by fusing the structural data and partial spectral data. These results indicate that the multiple linear regression model generated the more accurate yield estimation than the stepwise regression model. The independent variables removed from the stepwise regression model belonged to the VIs, and the final multiple linear regression equation determined the plant number, plant height, G, B, R/B, VARI, WI, and GLA as independent variables of the stepwise regression model.

## 4. Discussion

Some studies have shown that the spectral information obtained from UAV-based images has a certain potential for crop yield prediction, but its accuracy is far from satisfactory. In our study, there existed the following problems in using UAV-based images: (1) the spectral band provided was incomplete, since only visible spectrum information was used, (2) different crops had different sensitivities to the spectrum in different growth stages, and (3) the VIs were easily saturated in the dense vegetation [34], which would affect the accuracy and stability of the estimation results. Considering these, we propose to integrate spectral and structural information to estimate the yield of ramie.

### 4.1. Comparison of Field Measurements and UAV-Based Image Data

In recent years, great progress has been made in crop phenotype research based on computer vision technology, but it was unclear whether this technology could replace field measurements to monitor ramie growth. The UAV image-based plant height and plant number results were verified by the measured data (Figure 4).

The UAV-based plant height was found to be generally lower than the actual measured value, which might be attributed to the following: (1) the ramie canopy images captured by UAV remote-sensing system contained bare soil and other objects lower than the highest point of the plant. During the reconstruction of the canopy point cloud, the algorithm automatically removed the information of the small structure at the highest points, resulting in a lower plant height in the image than the actual plant height. (2) During the flight of the UAV, the high-speed rotation of the blade blew the ramie, resulting in some ramie toppling, eventually affecting the image plant height information.

The plant number is an important estimation index of the crop yield. The final yield of ramie can be better reflected by the plant number after the sealing period, since the ramie yield is affected by late tillering. In addition, compared with the plant counting in the early growth period, late plant counting is more challenging. This is because the canopy coverage and density increase as ramie grows, resulting in plant occlusion. Therefore, we only calculated the plant number on the sealing date. We found that some ramie plants were ignored and omitted by the YOLOv4 model, leading to a smaller UVA image-based plant number than the actual measured plant after counting. This might be due to the high canopy density of ramie, and in this case, the plant could not be effectively monitored.

### 4.2. Relationship between UAV-Based Image Data and Ramie Yield

Many studies have demonstrated that there is a significant correlation between spectral and structural features obtained by UAV-based images and crop yields [35,36,37,38]. Our study confirmed that the plant number, plant height, G, B, R/B, VARI, WI, and GLA obtained by UAV-based images could provide a good characterization of the ramie yield, but they possessed different potentials. Although spectral features can be used to reflect crop yields, there are still challenges about using a vegetation index to construct a yield estimation model. Not every vegetation index can accurately reflect the crop yield. If the correlation between the VIs and yield are unacceptable, the fusion of the VIs will result in a saturation of the spectral information and reduce the accuracy of the yield estimation model. We proved that the correlation between the WI and yield is the greatest. Therefore, the WI is one of the spectral variables of the ramie yield estimation model. The plant height and plant number can describe the ramie yield well, as shown in Table 3. This is because the plant height and plant number are the main factors affecting the ramie yield. In general, the ramie yield estimation model constructed by both spectral and structural features improved the accuracy of the yield estimation. In future research, other information can be considered to improve the reliability and accuracy of the model.

## 5. Conclusions

This study constructed ramie yield estimation models. We made the first attempt to estimate ramie yield by integrating UAV-based data. In this study, spectral and structural (plant number and plant height) information were extracted from UAV-based RGB images. The yield of the ramie was evaluated using a multiple linear regression model and a stepwise linear regression models. Our results showed that the UAV image-based plant height and plant number could reflect well the actual ground measured values, with a plant height R^2^ of 0.9095 and RMSE of 0.052 m and with a plant number R^2^ of 0.9647 and RMSE of 6.151, indicating that the structural information derived from the UAV images had high potential for the ramie yield estimations. However, our observations indicated that the spectral data obtained from the UAV images were less desirable for the ramie yield estimations. Our data also indicated that the yield estimation model we developed can predict a ramie yield more accurately and rapidly without damage to the crops, with an optimal R^2^ of 0.66 and the RMSE of 1.592 kg. Our study provided a reference for field production management.

## Figures and Tables

**Figure 1 sensors-21-00669-f001:**
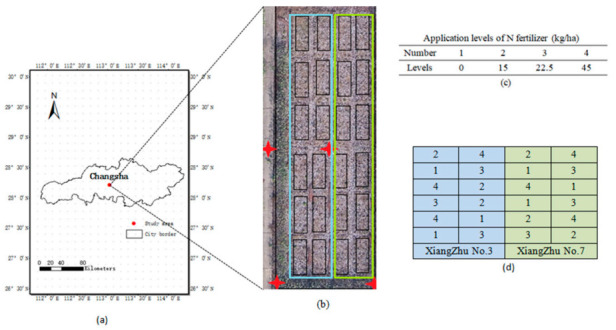
Study area and arrangement of the experimental sites: (**a**) geographic map of the study area, (**b**) experiment fields for ramie, (**c**) nitrogen fertilizer application levels represented by number, and (**d**) nitrogen fertilizer application levels in each plot.

**Figure 2 sensors-21-00669-f002:**
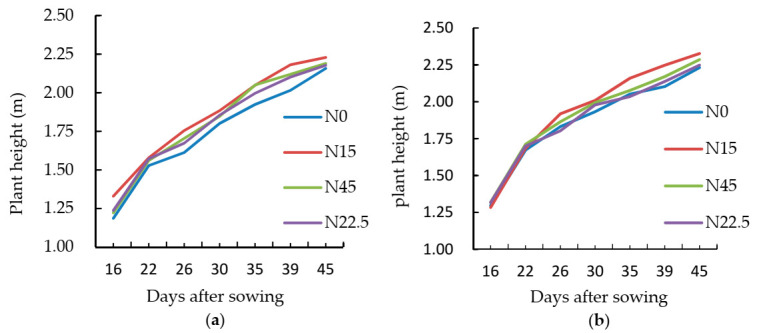
Variation trend of the digital surface model (DSM)-based plant heights. (**a**) XiangZhu No. 3 and (**b**) XiangZhu No. 7.

**Figure 3 sensors-21-00669-f003:**
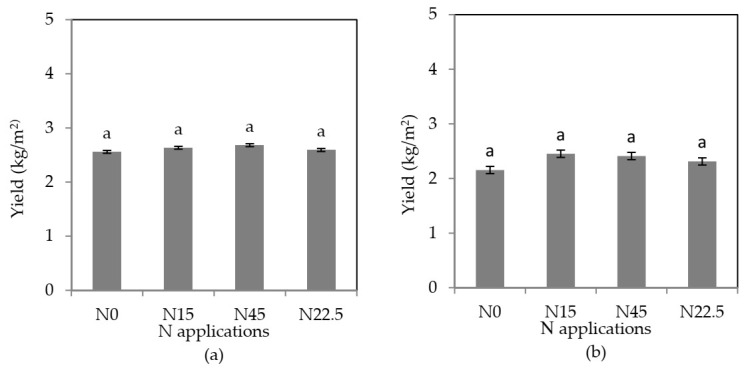
Effects of different nitrogen levels on the biomass of ramie. (**a**) XiangZhu No. 3 and (**b**) XiangZhu No. 7.

**Figure 4 sensors-21-00669-f004:**
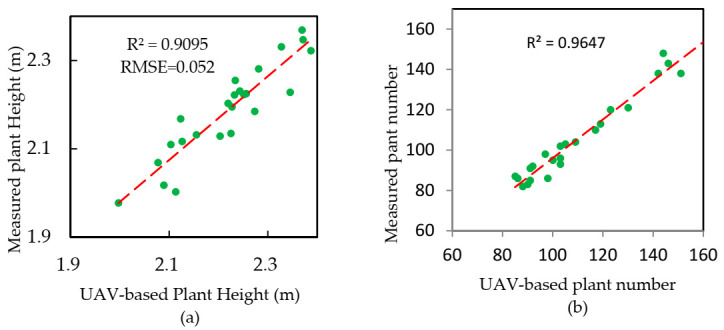
Precision analysis of the unmanned aerial vehicle (UAV)-based data. (**a**) Precision analysis of the UAV-based plant heights and (**b**) precision analysis of the UAV-based lant numbers.

**Figure 5 sensors-21-00669-f005:**
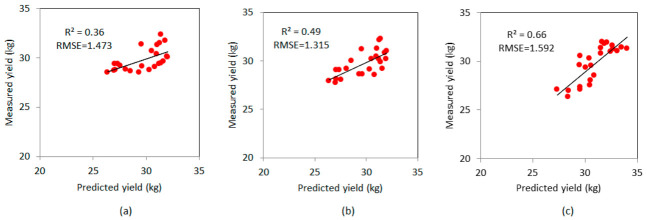
Estimation results of the ramie yield by using the multiple linear regression model and stepwise linear regression models. (**a**) Stepwise linear regression model only including the plant number, (**b**) stepwise linear regression model including the plant number and plant height, and (**c**) the multiple linear regression model.

**Table 1 sensors-21-00669-t001:** The flight details during the whole growth period.

Dates	Growth Stages	Number of Images Collected
2019/06/10	Seedling stage	55
2019/06/26	Seedling stage	55
2019/07/02	Sealing stage	55
2019/07/06	Sealing stage	55
2019/07/10	Prosperous long-term	55
2019/07/15	Prosperous long-term	55
2019/07/19	Prosperous long-term	55
2019/07/26	Mature period	55

Note: Seedling stage is when 50% of the ramie seeds are unearthed in the whole seedbed. Sealing stage means the period when the ramie canopy density increases, and there is no exposed land. Prosperous long-term is from the sealing date to the date when 1/3 of the stem blackened. Mature period is from the 1/3 of the stem blackened to the stem about 30 cm from the tip turned black.

**Table 2 sensors-21-00669-t002:** Indices calculated based on the unmanned aerial vehicle (UAV) images in this study.

Vegetation Indices	Reference
GLA = (2 × G−R−B)/(2 ×G + R + B)	[31]
ExR = 1.4R − G	[32]
ExG = 2 × G − R − B	[32]
ExGR = ExG − 1.4R − G	[32]
WI = (G − B)/(R − G)	[33]
NGRDI = (G – R)/(G + R)	[14]
RGBVI = (G × G – R × B)/(G × G + R × B)	[19]
VARI = (G − R)/(G + R − B)	[20]

Note: R represents the average pixel value of the red channel, G represents the average pixel value of the green channel, B refers to the average pixel value of the blue channel, and r, g, and b are the normalized values of R, G, and B.

**Table 3 sensors-21-00669-t003:** Correlation analysis of the UAV-based structural features and yield.

	Plant Number	Plant Height	Yield
Plant number	1		
Plant height	0.301	1	
Yield	0.600 **	0.524 *	1

Note: ** means a significant level of 0.01, and * means a significant level of 0.05.

**Table 4 sensors-21-00669-t004:** Correlation analysis of the UAV-based spectral features and yield.

	Yield	G/R	G/B	R/B	RGBVI	VARI	ExR	ExG	ExGR	WI	NGRDI	NDYI	GLA
Yield	1												
G/R	−0.06	1											
G/B	−0.331	0.536 **	1										
R/B	−0.36	0.268	0.957 **	1									
RGBVI	−0.292	0.704 **	0.974 **	0.870 **	1								
VARI	0.216	0.701 **	−0.221	−0.494 *	−0.012	1							
ExR	0.241	−0.840**	−0.908 **	−0.747 **	−0.976 **	−0.204	1						
ExG	−0.285	0.731 **	0.967 **	0.852 **	0.999 **	0.027	−0.984 **	1					
ExGR	−0.215	0.885 **	0.867 **	0.685 **	0.953 **	0.29	−0.996 **	0.965 **	1				
WI	0.365	−0.116	−0.889 **	−0.977 **	−0.785 **	0.626 **	0.631 **	−0.758 **	−0.561 **	1			
NGRDI	−0.052	1.000 **	0.531 **	0.262	0.700 **	0.705 **	−0.837 **	0.727 **	0.882 **	−0.11	1		
NDYI	−0.331	0.538 **	0.998 **	0.954 **	0.977 **	−0.223	−0.908 **	0.968 **	0.868 **	−0.896 **	0.533 **	1	
GLA	−0.284	0.730 **	0.966 **	0.852 **	0.999 **	0.025	−0.983 **	1.000 **	0.964 **	−0.761 **	0.726 **	0.969 **	1

Note: ** means a significant level of 0.01.

## Data Availability

Not applicable.

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
