# Peer review of "Ramie Yield Estimation Based on UAV RGB Images"

_sensors, 2021, doi:10.3390/s21020669_

Round 1

Reviewer 1 Report

Summary

The authors attempt to present methods and develop models to predict Ramie yield from plant counts and vegetation indices based on RGB images. The methods, but also the results presented in this manuscript seriously lack detail and other than the graphs showing the correlations between predicted and measured data, the reader is not given anything they could use to apply these methods in their own work. As such, the conclusion that RGB images can be used to successfully predict Ramie yield, is not very useful and not even really supported by the data that is presented. Not even the introduction is informative or sound as many of the references given do not seem to support the citations made.

Detailed comments

Line 11: “more or less destructive”… Please clarify. This is a vague statement.

Line 30: You reference Bendig et a. 2015 for the statement “Crop yield estimation is also required in market risk management and policy adjustment.” This paper does not touch on this topic at all. Please check you haven’t cited the wrong paper for this statement.

Line 33: Can you clarify what you mean by crop growth model here. It confuses me that you reference Zhang et al. 2016, because this paper is about crop identification and does not touch on crop modelling or yield predictions and the reference Schirrmann et al. 2016 only mentions it in one sentence.

Line 42: You make the statement that RGB cameras are becoming a promising tool, because they are cheap, but the title of one of the references you quote, Tian et al. 2016, tells me it is actually about hyperspectral and not RGB sensors.

Line 46: You say in your sentence “Chang et al. (2017) reported”, but then you give a different reference and Chang is missing in your bibliography.

Line 48: “Some sample vegetation indices such as VARI [20], NDYI [21], NGRDI  [22], and GLA [23] were calculated based on images acquired from the visible wavelengths.” What do you want to say with this statement? It seems a random collection of vegetation index references that don’t seem to have relevance to the research you are presenting in your paper. For example, the NDYI is a very specific index that has been used in canola with its yellow flowers.

Line 52: Here you talk about plant counts from UAVs compared to ground platforms, but the reference Zhang et al. 2020, does not seem to talk about such a comparison at all.

Line 382: Please check this reference in the bibliography. The last name should be Garcia-Martinez, not Hector.

Line 61: “Random forest…” not “Trandom forest regression..”

Line 86: Please give some more detail to the planting density or population density and the row spacings etc.

Line 87: Please spell out Nitrogen, the first time you are using it.

Line 89: Please give details of how the fertiliser was applied.

Fig. 1 d) is not needed.

Line 98: “an effective pixel number of 20.8 million”

Line 99: “During the whole growth period of second season of ramie…” It is not clear what you mean by that. When was your ramie crop planted? Please add more details.

Line 107: As someone not familiar with Ramie growing, I don’t know what you mean by prosperous long-term and sealing date. Unless they are commonly used terms in the Ramie community, please make sure you use the correct English translation or explain these terms. Also, what do you mean by shoot number and why is it

Line 115: The PIX4D mapper software was written by PIX4D in Lausanne, Switzerland. (PIX4d, Lausanne, Switzerland) (www. pix4d.com).

Line 126: Please spell out what HSV channel stands for.

Line 129: “wildly” > “widely”

Line 131: “In addition, some digital variables were also used to reflect the growth trend of ramie, including g/r, g/b, r/b.” It is not clear what you mean by growth trend and by those ratios. Please give references if they have been correlated with Ramie growth elsewhere.

Line 132: I could not find a reference to the GLA index in the Watanabe et al. 2017 reference that you give.

Nor could I find a reference to the ExR and ExG index in the Hunt et al 2005 reference and I have never heard of these indices either.

Similar with the rest of the indices you give. Please be sure you cite the correct papers and also give the wavelengths rather than just letters in your formulae.

Line 142: “ based on the field-measured plant height.” Please add to your method exactly how you measured height data in your field plots.

Line 145: Please give references for the YOLOv4 model.

Line 187, Fig. 2: Please give more detail about labels in your figure caption. E.g. spell out N treatments and what you mean by days growth? E.g. Days after sowing?

Fig. 3: Please give more details. E.g. is this averaged across the two varieties. Only show yield per m2.

Line 211-223: You will need to expand on this paragraph and show much more details of your results of the different analysis. This paragraph contains all your main objectives and results, but you are not really showing them in a way that they would support your conclusions.

Fig. 4: You will have to explain much better what each panel shows.

Line 217: You will need to give more detail in your statistical methods how you fused structural data with VI.

Line 227: You do not show any data or evidence for the 3 statements you are making here and your conclusion that it is superior to use structural and spectral information is not really supported by the data you present.

Line 239 and Fig. 5: These data belong into the results section.

Line 246: That constitutes a real problem with your method and you may have to fly higher than 12m to avoid this.

Line 249: What do you mean by heterologous germination?

Author Response

Thank you.

Reviewer 2 Report

From my point of view, this is a rather interesting manuscript that falls into the scope of Sensors scientific journal. Although the issue of estimating crop yields from RGB images has been known for decades, in the case of Ramie, I think it has not yet been addressed.

However, I found a number of shortcomings that should be addressed before publication.

p. 1, l. 22, 216, fig. 4, fig. 5: The coefficient of determination is given to 4 decimal places. But sometimes it is only to 2 decimal places (fig. 4). I think the article would benefit if R2 was always rounded to 2 decimal places.

p. 2, l. 61: Trandom forest regression – should be random forest regression.

p. 2, l. 69: there are few studies – please, be more concrete and add some reference.

p. 2, l. 88 and further: Why do you always use a different abbreviation (N or TB) for different doses of nitrogen? Wouldn't it be better to use a uniform abbreviation (for example, with the amount of nitrogen applied)? I mean N0, N15, N22 and N45….

p. 3, l. 108: Table 2 is not needed, it has only one row. The information from the table can be in one sentence of text.

p. 4, l. 129: wildly or widely?

p. 4, l.132, Table 3: Why didn't you also include the TGI (triangular greenness) index in the evaluation?

p. 6, l.189: Figure 3 shows…. not showed

p. 6, l. 195: certain amount of nitrogen fertilizer – please, be more concrete. What amount? Is it possible to do some qualified estimate?

p. 6, l. 196: The characters in parentheses are definitely not English.

p. 6, l. 209: Figure caption of Fig. 3. This also applies to some other figures in the text. All information written in the text in l. 212-214 should also be included in the figure caption. In general, a captioned image should be understandable without surrounding text.

p. 7, l. 227: there instead of therer

p. 8, l. 278: Moisture content term instead of humidity is used relating solid materials (as leaves are).

Author Response

Thank you.

Round 2

Reviewer 2 Report

Article was improved.

Author Response

Dear reviewer,

Thank you for your comments, it is very valuable to me.